# Non-aromatic annulene-based aggregation-induced emission system via aromaticity reversal process

Zheng Zhao[1,9], Xiaoyan Zheng [2,9], Lili Du [3,4,9], Yu Xiong[5], Wei He[1], Xiuxiu Gao[6], Chunli Li[6], Yingjie Liu[7], Bin Xu[7], Jing Zhang[1], Fengyan Song[1], Ying Yu[1], Xueqian Zhao[1], Yuanjing Cai [1], Xuewen He[1], Ryan T.K. Kwok[1], Jacky W.Y. Lam[1], Xuhui Huang [1], David Lee Phillips[3], Hua Wang [6] & Ben Zhong Tang [1,5,8]

Aggregation-induced emission (AIE) is a photophysical phenomenon correlated closely with the excited-state intramolecular motions. Although AIE has attracted increasing attention due to the significant applications in biomedical and optoelectronics, an in-depth understanding of the excited-state intramolecular motion has yet to be fully developed. Here we found the non-aromatic annulene derivative of cyclooctatetrathiophene shows typical AIE phenomenon in spite of its rotor-free structure. The underlying mechanism is investigated through photo-luminescence spectra, time-resolved absorption spectra, theoretical calculations, circular dichroism as well as by pressure-dependent fluorescent spectra etc., which indicate that the aromaticity reversal from ground state to the excited state serves as a driving force for inducing the excited-state intramolecular vibration, leading to the AIE phenomenon. Therefore, aromaticity reversal is demonstrated as a reliable strategy to develop vibrational AIE systems. This work also provides a new viewpoint to understand the excited-state intramolecular motion behavior of lumiongens.

[1] Department of Chemistry, Hong Kong Branch of Chinese National Engineering Research Center for Tissue Restoration and Reconstruction, Institute of Molecular Functional Materials, Division of Life Science and State Key Laboratory of Molecular Neuroscience, The Hong Kong University of Science and Technology, Clear Water Bay, Kowloon, Hong Kong 999077, China. [2] Beijing Key Laboratory of Photoelectronic/Electrophotonic Conversion Materials, Key Laboratory of Cluster Science of Ministry of Education, School of Chemistry and Chemical Engineering, Beijing Institute of Technology, 100081 Beijing, China. [3] Department of Chemistry, The University of Hong Kong, Pokfulam Road, Hong Kong 999077, China. [4] Institute of Life Sciences, Jiangsu University, 212013 Zhenjiang, China. [5] Guangdong Provincial Key Laboratory of Brain Science, Disease and Drug Development, Shenzhen Research Institute, No. 9 Yuexing 1st RD, South Area, Hi-tech Park, Nanshan, 518057 Shenzhen, China. [6] Engineering Research Center for Nanomaterials, Henan University, 475004 Kaifeng, China. [7] State Key Laboratory of Supramolecular Structure and Materials Jilin University, 130012 Changchun, China. [8] NSFC Center for luminescence from Molecular Aggregates, SCUT-HKUST Joint Research Institute, State Key Laboratory of Luminescent Materials and Devices, South China University of Technology, 510640 Guangzhou, China. [9] These authors contributed equally: Zheng Zhao, Xiaoyan Zheng, Lili Du. Correspondence and requests for materials should be addressed to D.L.P. (email: phillips@hku.hk) or to H.W. (email: hwang@henu.edu.cn) or to B.Z.T. (email: tangbenz@ust.hk)

The utilization of microscopic molecular motion to design macroscopic materials is receiving gradually more attention as it not only affords versatile functional materials but also broadens the fundamental understanding of dynamic molecular systems[1,2]. In this regard, scientists have made great achievements with fascinating concepts and materials such as in making molecular machines (e.g., molecular motors, molecular switches, and molecular shuttles), stimuli-responsive polymers and smart fluorescent materials etc. were all well proposed and developed[3–10]. Among them, molecular motion responsive fluorescent materials have attracted much attention due to their wide applications in biological systems, materials science and membrane chemistry[7–11]. While unlike the other molecular motion responsive systems such as molecular motors that rely on controlling and promoting the molecular motion to attain specific functionality, fluorescent systems generally need the suppression of molecular motion to generate a strong fluorescence[12,13]. A concept that has attracted increasing attention in this context is the development of aggregation-induced emission (AIE)[12]. Typical AIE systems show quenched emission in the solution state since the dynamic intramolecular motion accelerates the non-radiative decay process of the excited state[12–15], while in the solid state the restriction of intramolecular rotation (RIR) and intramolecular vibration (RIV) suppresses the non-radiative decay and thus generate a high-fluorescence quantum efficiency[12]. So far, AIE systems based on RIR have been deeply investigated both mechanistically and in practice, based on which versatile solid-state fluorescent materials and stimuli-responsive fluorescent probes were developed[12–16]. However, molecular systems based on RIV are still rather limited in variety due to the unclear molecular design and the intrinsic difficulty in regulating the vibrational magnitude than the rotation magnitude, although vibrations are more general than rotation as a motion mode[17,18]. Additionally, an in-depth understanding of the excited-state molecular motion behavior and elucidation of the origin of the molecular motion are yet to be explored.

In general, most of the developed luminogens are based on aromatic compounds since they hold better electron conjugation and are more stable thermodynamically than the appropriate non-aromatic systems[19–23]. However, aromatic compounds generally show suppressed molecular motion via superior electron conjugation, which makes their vibrational/rotational degrees of freedom and torsional space rather limited for further tuning. In this regard, the annulene system with non-aromaticity shows a large magnitude of vibration upon photoexcitation, which provides a suitable platform for studying the influence of vibrational behavior on the exciton relaxation[24–26]. According to the aromaticity reversal theory proposed by Baird in 1972 (Baird's rule)[27–30], the aromaticity (antiaromaticity) of annulenes in the triplet excited-state ($T_1$) is reversed to that in the ground state. In other words, annulenes with $[4n + 2]$ π electrons show aromatic character in the ground state but antiaromatic character in the excited state. Similarly, annulenes with $[4n]$ π electrons exhibit antiaromatic character in the ground state but aromatic feature in the excited state. This theory later has been found also applicable to the singlet excited state ($S_1$)[31–37]. Since the aromaticity reversal will result in a change of molecular conformation, the excited-state aromaticity reversal can possibly serve as a driving force to induce a large amplitude conformation flip to quench the emission and the suppression of which theoretically can light up the emission, leading to the generation of a new type of AIE system.

Cyclooctotetraene (COT) is a prototype molecule of non-aromatic annulenes[32–35]. Recently, versatile aromatics/hetero-aromatics fused COT systems have been developed with some of them are suitable candidates to explore excited-state aromaticity reversal[29,38–45]. For instance, Itoh and colleagues[29] energetically quantified Baird's aromaticity via investigation of the inversion of photoexcited cyclooctatetrathiophene (COTh) derivatives. Hada et al. monitored the dynamic structural change of thiazole fused COT with liquid crystal character induced by excited-state aromaticity reversal[38]. Although these COT derivatives show active conformation flip after photoexcitation, their luminescent properties have not been investigated, which inspired us to explore the possibility of using COT derivatives to construct vibrational AIE system and unveil the relationship among the aromaticity reversal, molecular motion, and luminescent properties.

In this work, we investigate the luminescent behavior of COTh and its derivatives by photoluminescence (PL) spectra, time-resolved absorption spectra, theoretical calculations, circular dichroism (CD), and circular polarized luminescence (CPL) spectroscopy, as well as pressure-dependent fluorescent spectra. COThs are demonstrated to be a unique vibrational AIE system (Fig. 1). Furthermore, the relationship between the AIE property and the excited-state aromaticity reversal is elucidated. The results here indicate that the aromaticity reversal from ground state to the excited state serves as a driving force for the structural vibrational behavior of COThs, leading to their AIE phenomenon. Furthermore, because of the chirality and the enhanced emission of COThs in aggregates and solid state, some of its derivatives exhibit CPL and microbial imaging capability. Aromaticity reversal thus is demonstrated as a reliable strategy to develop new AIE systems. Furthermore, the extensive application platform of AIE materials also provides a new outlet for utilizing Baird's rule to develop luminescent functional materials.

## Results and discussion

**Molecular synthesis and characterization**. The molecular structures involved have been presented in Fig. 2. Their synthetic routes are shown in the Supplementary Figs. 1 and 2. In general, COTh and COTh-tetramethylsilane (TMS) and COTh-Ph were synthesized according to literature methods[46,47]. COTh-Py was synthesized by using a $S_{N2}$ reaction of **1** and pyridine at room temperature that gave a yield of 57%. While **1** was produced through the $S_{N2}$ reaction between **2** and 1,6-dibromohexane (yield: 52%). The demethoxy derivative of **3** can generate **2** in a yield of 62%. **3** was synthesized by Suzuki coupling between brominated COTh and 4-methoxy phenylboronic acid efficiently with a yield of 70%. The new compounds have been characterized by $^1H$ NMR, $^{13}C$ NMR, and high-resolution mass spectroscopies with satisfied results (Supplementary Figs. 2–13).

**Photophysical properties**. The optical properties of COTh and COTh-TMS were first investigated using ultraviolet (UV)–visible (Vis) and photoluminescence (PL) spectra (Fig. 3 and Supplementary Figs. 14–16). COTh showed a strong absorption peak at 280 nm and a weak shoulder peak at around 340 nm, which could be assigned to a ππ* and nπ* transition, respectively. COTh-TMS shows slightly redder absorption wavelengths than that of COTh (310 nm, shoulder: 350 nm) due to the TMS substitution. In the film state, both the absorption of COTh and COTh-TMS show a slight red-shift, possibly due to the enhanced intermolecular interaction and/or molecular planarization. Interestingly, COTh-TMS exhibited AIE activity with dim emission in dilute solution but bright green emission in the solid state ($\Phi_F$: photoluminescence quantum yield, $\Phi_{F, soln.} = 0.7\%$, $\Phi_{F, solid} = 10\%$, determined by using a calibrated integrating sphere) as shown in Fig. 3. The AIE characteristic was also confirmed by studying its PL behaviors in acetone/water mixtures with different water fractions ($f_w$) (Supplementary Fig. 15). At a low $f_w$, the mixture exhibited negligible PL emission, while at a high $f_w$, the PL intensities became very strong due to the formation of aggregates.

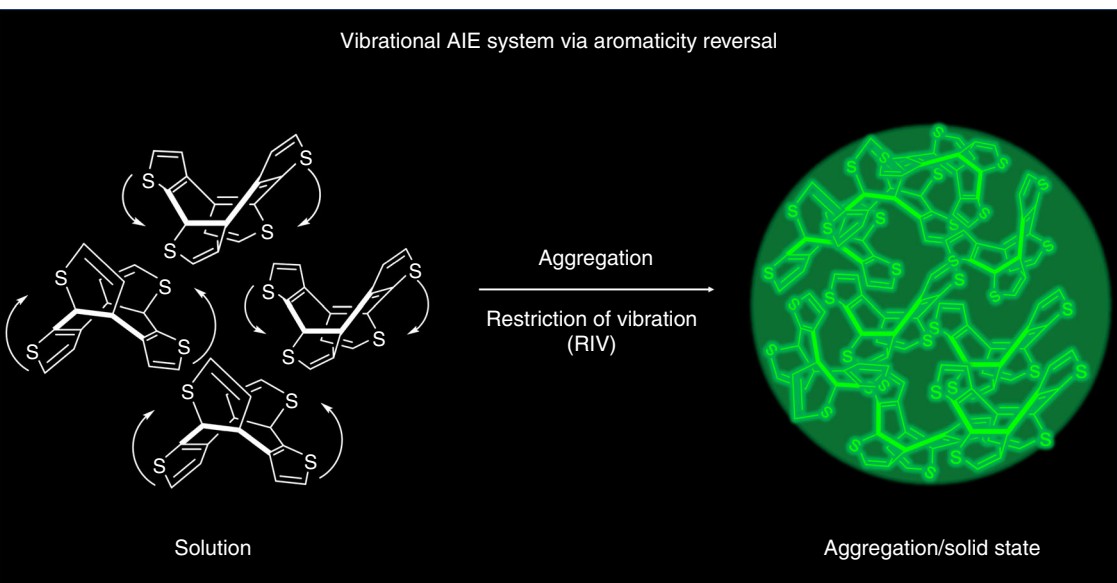

**Fig. 1** Schematic diagram illustrating the AIE process of COTh system. In solution, the active vibration of COTh upon excitation quench the emission, while in the aggregates or solid state, the restriction of vibration light up the bright green emission

According to the RIM mechanism proposed by Tang and co-workers[12], the non-emissive characteristic of COTh-TMS possibly is ascribed to its active intramolecular motion in the solution state. We thus measured its PL spectra at room temperature and 77 k because the molecular motion can be greatly suppressed at 77 k (Supplementary Fig. 16). COTh-TMS is dimly emissive at room temperature but brightly emissive at 77 k, suggesting that molecular motion possibly plays a key role for the AIE phenomenon. Initially, we assume that the TMS group possibly works as rotors to promote the non-radiative decay process, resulting in the AIE phenomenon of COTh-TMS. However, further investigation indicates COTh, an analog of COTh-TMS without TMS group also shows a typical AIE property ($\Phi_{F, soln,}$ = 0.4%, $\Phi_{F, solid,}$ = 11%). This suggests that the molecular backbone vibration rather than the rotation of the TMS group plays a predominant role for the AIE phenomenon of the COTh system, although the TMS rotation may contribute to the increased AIE effect of COTh-TMS with a larger $\alpha_{AIE}$ value than that of COTh (Supplementary Fig. 15).

**Single crystal analysis and theoretical calculation.** To gain insight into the details of the AIE phenomenon of COTh and its derivatives, we analyzed their crystal structures (Supplementary Figs. 17−19)[46,47]. COTh and its derivatives examined all showed non-coplanar and saddle-type conformations, which avoid strong intermolecular π–π interactions that are detrimental for the luminescence to take place[12]. Furthermore, multiple intermolecular C−H…π and S…π interactions exist in the crystals, which help to rigidify the molecular conformation and are beneficial for the solid state emission to occur[12,18]. It is worthy to note that the non-coplanar and saddle-type conformations of COTh are similar to that of COT, which implys vibrational molecular motion possibly occur for COTh upon photoexcitation due to lack of aromaticity stabilization[32–34]. To better understand the influence of the molecular motion on the AIE phenomenon of COTh, theoretical calculations were carried out for the ground state ($S_0$) at the DFT/B3LYP/SV(P) level and for the first excited singlet state ($S_1$) at TDDFT/B3LYP/SV(P) level, respectively, for the COTh molecule in both the gas phase and in its crystal state (Table 1, Fig. 4, Supplementary Figs. 20−21, and Supplementary

Table 1−4)[48]. In Table 1 and Supplementary Table 1, we present selected major geometrical stuctural parameters of the ground ($S_0$) and singlet excited state ($S_1$) for the COTh molecule, as well as the corresponding X-ray crystal structure data for comparison. The optimized geometry of COTh at the $S_0$ minimum in the solid state is consistent with the crystal structure. This confirms the reliability of the QM/MM method at the B3LYP/SV(P) level and the general Amber force field[49]. It is clear that the isolated COTh molecule is more flexible than the molecule in a cluster, and this is reflected by the structral differences between $S_0$ and $S_1$ for the representative dihedral angles in Table 1. The torsional angles $\Theta_{I–II}$ and $\Theta_{II–III}$ are 22.82° and 24.22° in the gas phase, far larger than those in the solid state of 12.78° and 12.96°. Additionally, the torsional angle between the neighboring thienyl rings are decreased in the excited state, indicating a better planarity of the central large ring V. These results suggest that the four wings of COTh undergoes an up-down molecular motion from $S_0$ to $S_1$, supporting the existence of a vibrational motion behavior. The electron density contours of HOMO and LUMO of COTh at their side and top views are shown in Fig. 4b and Supplementary Fig. 20, respectively. The HOMO is of π character, while LUMO is of π* character. Both the HOMO and LUMO delocalized in the whole molecule. In addition, the transition between the HOMO and LUMO plays a significant contribution (>99%) for the singlet excited state (Supplementary Table 2). The AIE effect of the COTh molecule was explicitly confirmed by theoretically calculation results for the radiative rate ($k_r$), the non-radiative rate ($k_{ic}$) and the fluorescence quantum yield ($\eta_F$), (Supplementary Table 3). It was demonstrated that the fluorescence quantum yield of COTh in the solid state is three orders of magnitude larger than that in the isolated state, because of the significant enhancement of $k_r$ and decrease of $k_{ic}$. The increase of the $k_r$ in the solid state should be attributed to the larger oscillator strength (f) and larger adiabatic excitation energy (Supplementary Table 3). More importantly, the $k_{ic}$ of COTh in the solid state was two orders of magnitude smaller than that of the isolated molecule. The reorganization energy of the solid state is much smaller than that in the gas phase and is a key factor influencing the value of the $k_{ic}$. We further projected the reorganization energy into the internal coordinates to explicitly partition the total reorganization

**Fig. 2** Structures and synthetic routes. **a** Structure of COTh and its derivatives. **b** The synthetic route to COTh-Py

energy into bond lengths, bond angles and dihedral angles of the COTh molecule in both the gas phase and in the solid state. As shown in Supplementary Table 4, the major differences in the reorganization energies between the gas phase and the solid state should be due to the dihedral angles, which corresponds to the low-frequency normal modes (Fig. 4c). Thus, the low-frequency vibrational motion plays a significant role for the AIE phenomenon of COTh system.

**The excited-state conformation reversal.** Circular dichroism (CD) spectroscopy can be used to monitor the stereo conformation change of a chiral compound[50,51]. Since COTh is conformationally chiral, we did the chiral resolution of *rac*-COTh via preparative chiral HPLC (Supplementary Figs. 22−24) and investigated the dynamical change of the chiral properties in the excited state using CD spectroscopy (Fig. 5 and Supplementary Figs. 25−27). It is worthy to note that this experiment is exact the same measurements performed previously by Itoh and colleagues, in which they investigated the conformation change of 2-methylthiphene fused COT, a structural ananlogue of COTh,

via CD spectroscopy. The stereo conformations of the two enantiomers were confirmed by their single crystal structures (Supplementary Table 5, Table 6, and Supplementary Fig. 25). The two enantiomers show strong CD signals, which are mirror symmetrical. Interestingly, the CD signals of the two enantiomers in freshly distilled tetrahydrofuran (THF) solutions gradually decreased upon UV irradiation and totally racemized after 14 min UV irradiation. As a contrast, without UV irradiation, the CD signals are almost unchanged. These results indicate that in the solution state, the two enantiomers of COTh can transform with each other through ring inversion upon excitation, which is consistent with its up-down molecular vibrational behavior. We then measured their CD spectra in the solid state upon UV irradiation under the same conditions with same irradiation time and the results indicate that the CD signals show almost no change, suggesting that the conformation inversion was suppressed in the solid state. The CD data suggest that the up-down conformation inversion behavior is the main molecular motion active for COThs, which should play a key role for the AIE phenomenon. Based on the chiral property and enhanced PL of COTh in the solid state, we then explored the CPL property of the

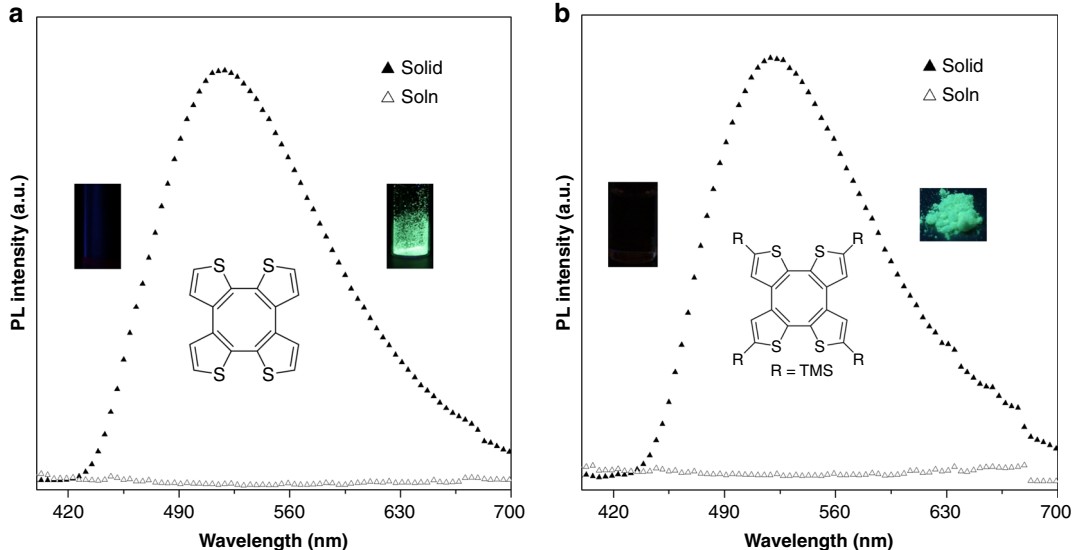

**Fig. 3** Photophysical properties. **a** The emission of THF solution (empty triangles) and solid (full triangles) of COTh, concentration: 10 μM, excitation wavelength is 350 nm. Inset: the fluorescence pictures of THF solution (left, dark) and solid (right, green fluorescence) of COTh taken under an excitation wavelength of 365 nm by a portable UV lamp. **b** The emission of THF solution (empty triangles) and solid (full triangles) of COTh-TMS, concentration: 10 μM, excitation wavelength is 350 nm. Inset: the fluorescence pictures of THF solution (left, dark) and solid (right, green fluorescence) of COTh-TMS taken under an excitation wavelength of 365 nm by a portable UV lamp

**Table 1 Calculation results of COTh in both the gas phase and the crystal phase**

| | Gas | | | Solid | | | Crystal |
|---|---|---|---|---|---|---|---|
| | $S_0$ | $S_1$ | $|\triangle S_0 - S_1|$ | $S_0$ | $S_1$ | $|\triangle S_0 - S_1|$ | |
| $C_3$-$C_2$-$C_1$-$C_8$ ($\Phi_I$) | 0.25 | 1.37 | 1.12 | 1.99 | 5.272 | 3.28 | 1.20 |
| $C_5$-$C_4$-$C_3$-$C_2$ ($\Phi_{II}$) | 0.24 | 1.37 | 1.13 | −0.44 | −1.65 | 1.21 | 0.83 |
| $C_7$-$C_6$-$C_5$-$C_4$ ($\Phi_{III}$) | 0.25 | 1.37 | 1.12 | 2.16 | 5.94 | 3.78 | 4.15 |
| $C_1$-$C_8$-$C_7$-$C_6$ ($\Phi_{IV}$) | 0.24 | 1.37 | 1.13 | −0.58 | −2.738 | 2.16 | −1.2 |
| $C_{12}$-$C_3$-$C_2$-$C_{11}$ ($\Theta_{I-II}$) | 44.93 | 22.11 | 22.82 | 43.84 | 31.06 | 12.78 | 42.85 |
| $S_{15}$-$C_5$-$C_4$-$S_{14}$ ($\Theta_{II-III}$) | −48.58 | −24.36 | 24.22 | −48.67 | −35.71 | 12.96 | −48.83 |
| $C_{18}$-$C_7$-$C_6$-$C_{17}$ ($\Theta_{III-IV}$) | 44.92 | 22.11 | 22.81 | 45.24 | 32.6 | 12.64 | 45.16 |
| $S_9$-$C_1$-$C_8$-$S_{20}$ ($\Theta_{I-IV}$) | −48.59 | −24.36 | 24.23 | −48.94 | −35.72 | 13.22 | −46.99 |

crystal of the enantiomers of COTh, and both of the enantiomers exhibit obvious CPL signals with the $g_{CPL}$ values around $10^{-3}$, and the CPL signals of the crystals also show almost no change under same UV irradiation as the CD experiments, which should be ascribed the restricted conformation inversion in the solid state (Supplementary Figs. 28 and 29).

**Aromaticity reversal**. In comparison with typical AIEgens that generally show intramolecular rotation behavior, the large magnitude of the up-down conformation inversion of COTh molecules is kind of unique. According to a previous study about Baird's aromaticity, the excited-state aromaticity reversal can lead to this large magnitude of up-down conformation inversion[25,26]. Indeed, the pioneering work by Itoh and colleagues[29] and Hada et al.[38] indicated methyl substituted COTh derivatives underwent a large magnitude of up-down conformation inversion upon photoexcitation. Therefore, the up-down ring inversion of COTh is ascribed to the aromaticity reversal from the ground state to excited state. By analyzing the geometric structures at the minima and at transition state of COTh in both the ground and excited states via Gaussian 09 at the (TD)DFT/B3LYP/6−31G** level, we indeed found that, in contrast to the ground state, both the structures at the global minima and transition states on potential energy surface of the excited state of COTh exhibited a more

planar conformation and a diminished bond length difference between the single and double bonds, which matched well with its aromatic character at the excited state (Supplementary Figs. 30 and 31). Furthermore, nucleus-independent chemical shift (NICS) and anisotropy of the induced current density (ACID) are commonly utilized methods to evaluate the aromaticity of the molecules, negative NICS value, and clockwise ring current usually suggest the aromaticity while positive NICS value and counterclockwise ring current indicate the antiaromaticity. Here, we also performed the NICS the ACID calculations, the $NICS_{zz}$ scan of COTh, perpendicular to its central 8-member ring, demonstrate positive and negative minima for the transition states at $S_0$ and $T_1$ states respectively, which is consistent with the ground-state antiaromaticity and excited-state aromaticity characteristic of COTh (Supplementary Fig. 32). Furthermore, ACID for the transition state structures in the $S_0$ and $T_1$ states show counterclockwise and clockwise ring current, which also supports the ground-state antiaromaticity and excited-state aromaticity characteristic of COTh (Supplementary Fig. 33).

**Transient-absorption spectroscopy**. Ultrafast optical spectroscopy gives a wealth of information on the photophysical properties and dynamical processes of the excited state. This enables us to investigate the excited state relaxation behavior of

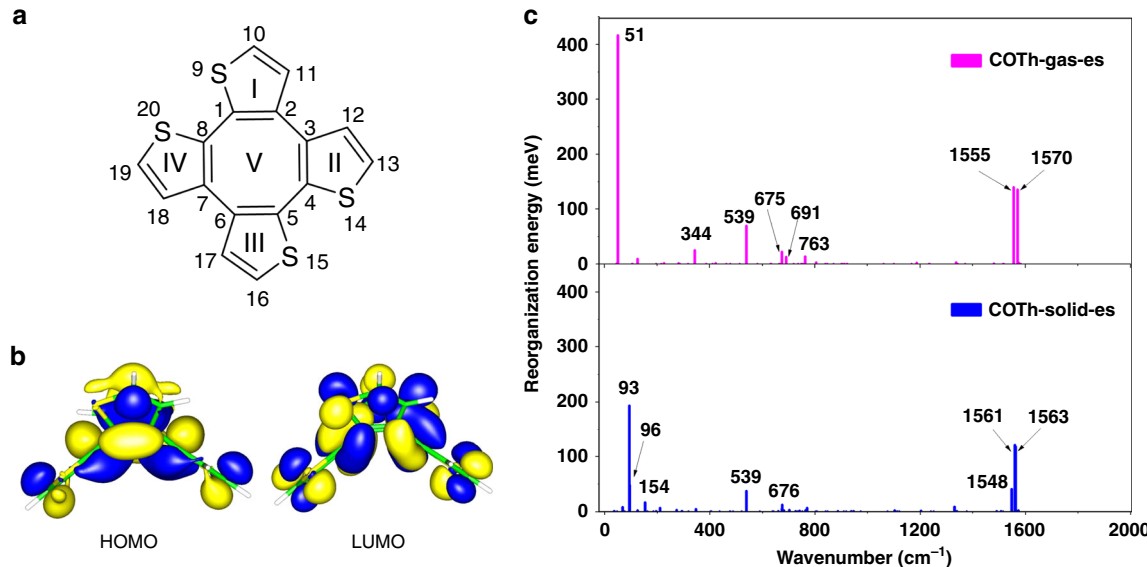

**Fig. 4** Theoretical calculations. **a** The chemical structure of COTh, with both atoms in the backbone and five rings labeled, respectively. **b** The electron density distributions of the molecular orbitals HOMO and LUMO of COTh molecule at the side view. **c** Plots of the calculated reorganization energies vs. the normal mode wavenumbers of COTh for the gas phase (upper panel) and the crystal structure (lower panel)

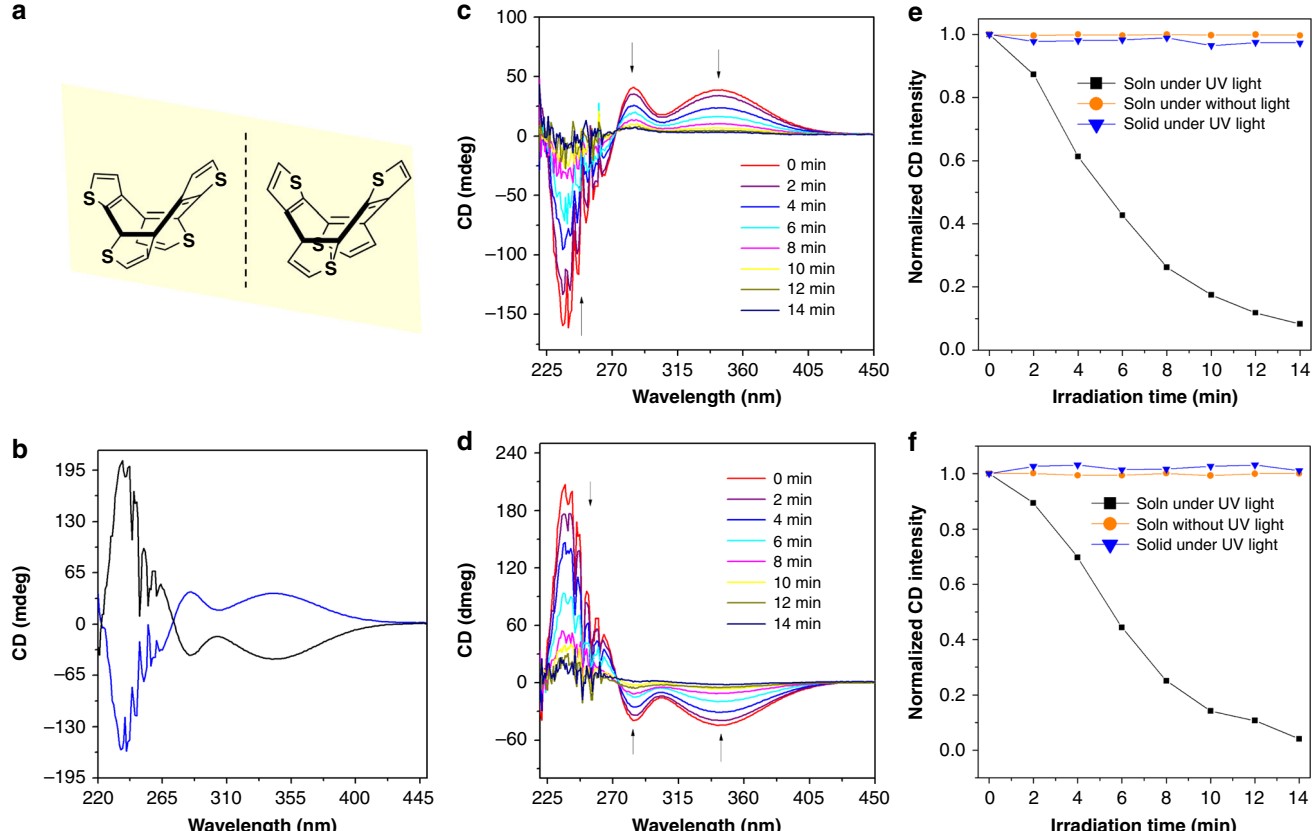

**Fig. 5** Circular dichroism spectroscopy. **a** The two enantiomers of COTh. **b** CD spectra of the two enantiomers of COTh in freshly distilled THF solution at 298 k. **c**, **d** Time-dependent CD spectral change profiles of the two enantiomers of COTh in freshly distilled THF solution under UV lamp irradiation at 298 k. **e**, **f** Time-dependent CD spectral change profiles of the two enantiomers of COTh in freshly distilled THF solution with UV lamp irradiation (black square line) and without UV lamp irradiation (orange circle line), and in the solid state with UV lamp irradiation (blue triangle line) at 298 k, irradiation wavelength: 254 nm. Concentration of the two enantiomers of COTh (−) and COTh (+) are 30 and 50 μM, respectively

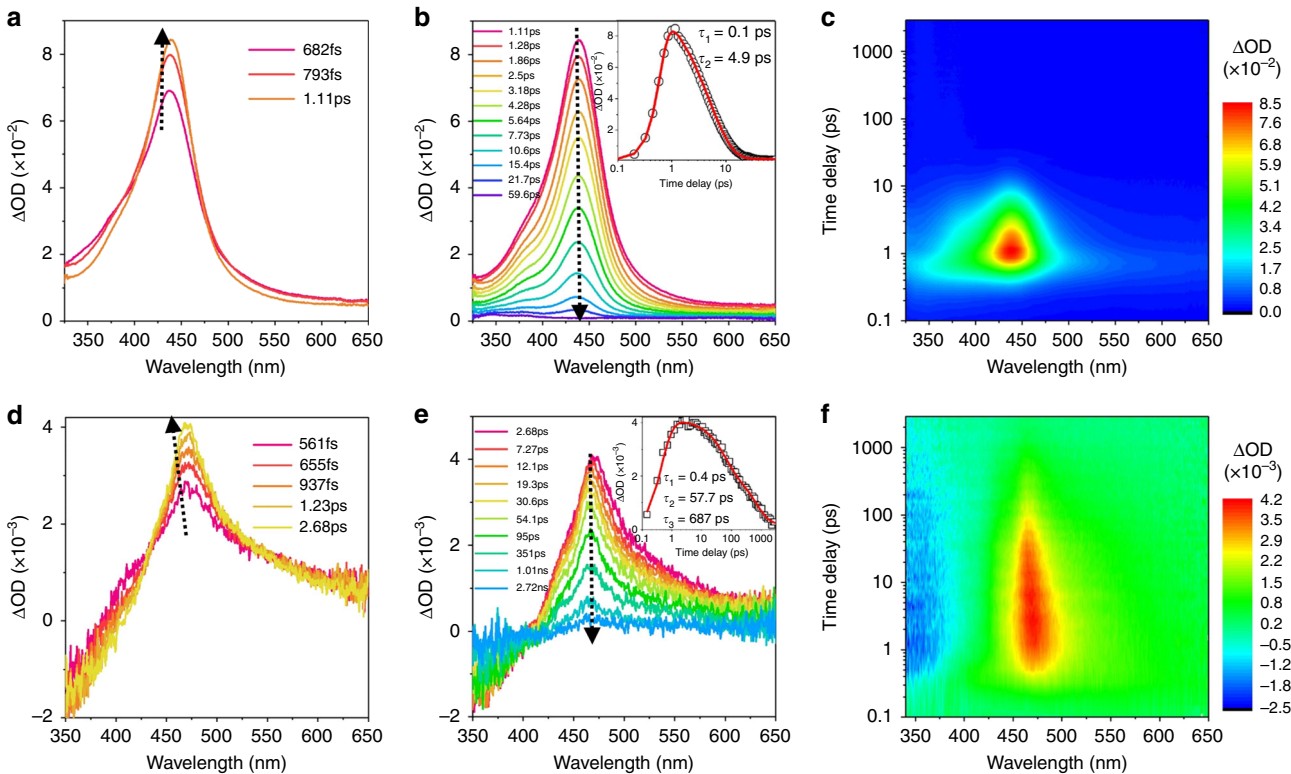

**Fig. 6** Excited-state dynamics by ultrafast time-resolved spectroscopy. **a**, **b** Femtosecond transient-absorption (fs-TA) spectra of COTh in MeCN at different time delays acquired after femtosecond laser excitation at 267 nm. Inset: kinetic traces at 440 nm (black circle) are inserted in **b**. **d**, **e** fs-TA spectra of COTh film at different time delays acquired after femtosecond laser excitation at 267 nm. Inset: kinetic traces at 470 nm (black circle) are inserted in **e**. **c** and **f** Contour plots of the time-resolved absorption spectroscopic responses are shown. $\Delta OD = \log \frac{I_{100}}{I_T(t,\lambda)}$, $I_{100}$ is the light level measured through the sample before excited states are created, $I_T$ is the transmitted light through the sample

COTh[52,53]. The femtosecond transient-absorption (fs-TA) spectra of COTh obtained after 267 nm excitation in acetonitrile with a time delay up to 59 ps are shown in Fig. 6a−c. As presented in Fig. 6a, an absorption band at 450 nm emerged whose intensity increases with a delay time from 682 fs to 1.1 ps. In the later delay times, the band at 450 nm decays from 1.1 ps to 59 ps. The kinetics at 450 nm could be fitted satisfactorily by a two-exponential function with time constants of 0.1 ps and 4.9 ps, respectively. The first time constant (0.1 ps) originates from the internal conversion (IC) process from $S_n$ to $S_1$ while the second time constant (4.9 ps) is assigned to the IC process from $S_1$ to $S_0$, corresponding to the ring inversion process. Therefore, the ring inversion process is very rapid. In the thin film state, a similar transition from $S_n$ to $S_1$ and $S_1$ to $S_0$ were observed. However, the kinetics at 470 nm affords three time constants, with two shorter ones (0.4 ps and 57.7 ps) corresponding to the IC process and a longer one (687 ps), which suggests a radiative decay. Therefore, both non-radiative decay and radiative decay processes coexist in the film state, which is in accordance with its not very high-fluorescence quantum yield. These data suggest that COTh has a very rapid molecular deformation in the solution state, resulting in a non-emissive property. While in the solid state, the molecular deformation was partially suppressed, which leads to the observed enhanced emission.

**Proposed sketch of the AIE mechanism of COTh**. To illustrate the AIE mechanism of COTh and the relationship between the AIE phenomenon and aromaticity reversal, a hypothetical dia-grammatic sketch that fits the data is shown in Supplementary Fig. 34. Supplementary Table 7 gives the criteria to judge the involved concepts such as aromaticity, antiaromaticity, and non-

aromaticity. In general, the chiral COTh in the solution state adopts a tub-like conformation in the ground state due to its non-aromatic characteristic, which will exceed a quick molecular deformation after excitation to approach the planar/quasi-planar aromatic state (transition state) according to Baird's rule. The planar/quasi-planar excited state relaxed to the ground state within picosecond timescale, in which the molecular conformation is almost unchanged. As the planar conformation of COTh in the ground state is antiaromatic, it will further relax to the non-aromatic tub-like conformation quickly to avoid the destabiliza-tion of antiaromaticity. The molecular deformation of the planar excited state has two pathways, e.g., go up and go down, conse-quently, the racemization occurred. It is also worthy to note that the racemization also possibly occurs in the excited state, where the two minimum energy structures (MES) adjacent the transi-tion state can overcome the energy barrier to transform with each other and result in the racemization. In that case, the exciton relaxation possibly proceeds via the MES besides the planar transition state. No matter in which pathway, the whole con-formation flip and exciton relaxation process occur within pico-seconds, corresponding to the non-radiative decay of the exciton, the emission was thus quenched (Supplementary Fig. 34a). However, in the solid state, the energy barrier for conformation change is large enough to restrict the molecular deformation process, the radiative decay pathway was thus opened, resulting in the enhanced emission in the solid state (Supplementary Fig. 34b).

**Pressure-dependent fluorescent spectroscopy**. As the molecular deformation of the luminogen correlates with their luminescence behavior, the luminescence of COTh upon hydrostatic pressure

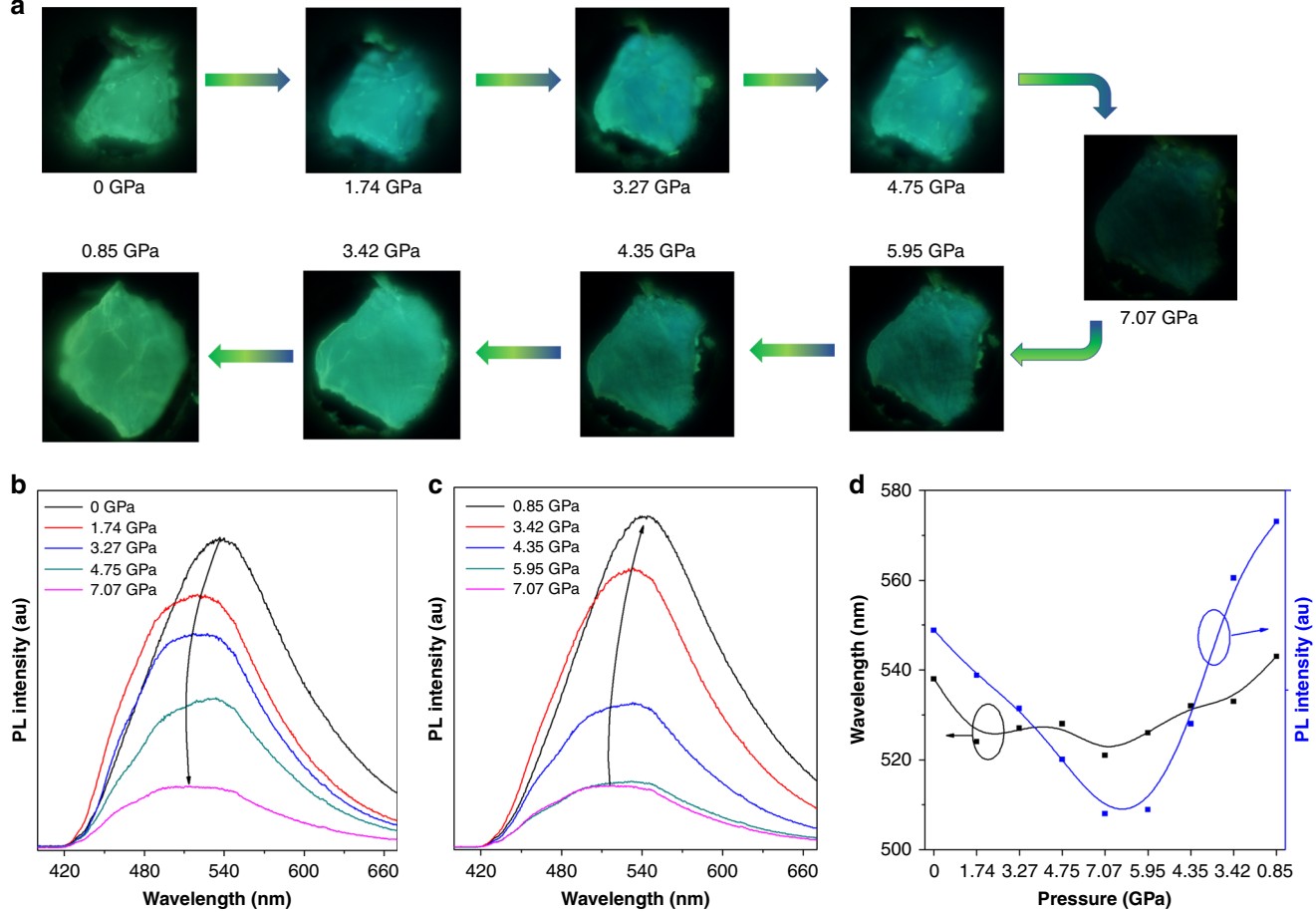

**Fig. 7** Fluorescence changes upon varied hydrostatic pressure. **a** Micrographs of the crystal under high pressure. **b** Emission spectra recorded during the compression process in the range of 0−7.07 GPa. **c** Emission spectra recorded the decompression process in the range of 7.07−0.85 GPa. **d** Plot of the emission wavelength (black dot line) and PL intensity (blue dot line) vs. the hydrostatic pressure

was thus investigated by using diamond anvil cell (DAC) equipment (Fig. 7)[54,55]. Indeed, with the external pressure gradually increased from 0 atm to 7 GPa, the emission color and spectra of the crystal underwent a blue-shift with the emission intensity becoming dimmer. Upon releasing the pressure to initial state, the emission could almost completely recover to the original position, corresponding to the molecular deformation and recovery. Geometry optimization upon varying pressures indicates that under high pressure, the intermolecular distance become smaller and the molecular conformation tends to more twisted (Supplementary Figs. 35 and 36). The smaller intermolecular distance suggests stronger intermolecular interaction, which caused the quenching of the emission under high pressure, while the more twisted structure should be responsible for the emission to blue-shift.

**Bio-imaging application**. To confirm whether the AIE character of COTh could be extended to other COTh systems, we also synthesized other derivatives by phenyl and methoxyphenyl substitution and all of the prepared derivatives show AIE activity (Supplementary Figs. 37 and 38). In particular, COTh-Py is water-soluble with weak emission in water solution but bright yellow emission in its aggregation state, enabling it to become a suitable candidate for a wash-free bio-imaging application. We further explored the microbial imaging ability of COTh-Py since its positive charge is inclined to interact with the negative charge on the surface of microbial, which would restrict the vibration of

the four arms of COTh-Py and thus light up the emission[56]. As shown in Supplementary Fig. 39, COTh-Py can stain *E. coli* and *P. chrysogenum* by a wash-free method with low background, suggesting its great potential in microbial detection.

## Discussion

COThs, a new family of non-aromatic annulene-based luminescent molecules has been developed and demonstrated to be a unique vibrational AIE system. The aromaticity reversal triggered intramolecular motion has been demonstrated experimentally and theoretically to play a key role for the AIE phenomenon of COTh system. This new AIE system not only enriched the AIE family, but also provided a reliable strategy for the design of vibrational AIE systems. Additionally, based on the chiral and AIE properties of the COTh system, CPL and microbial imaging application were explored. To the best of our knowledge, this is the first time that the AIE phenomenon has been considered from the viewpoint of aromaticity inversion, which fundamentally helps understanding the excited-state relaxation of luminescent molecules. Furthermore, the extensive application platform of AIE materials also provides a new outlet for utilizing Barid's rule to develop luminescent functional materials.

## Methods
**General**. All chemicals were commercially available and used as supplied without further purification. Deuterated solvents were purchased from J&K. 3,3′-bithio-phene was purchased from Derthon Optoelectronic Materials Science Technology

Co., Ltd. Tetrahydrofuran (THF) was dried by distillation using sodium as drying agent and benzophenone as indicator. The $^1$H and $^{13}$C NMR spectra were recorded on a Bruker ARX 400 MHz and Bruker Biospin GmbH 600 MHz NMR spectrometer using the deuterated solvent as the lock and TMS ($\delta = 0$) as internal reference. Mass spectra and High-resolution mass spectra (HRMS) were obtained on a Finnigan MAT TSQ 7000 Mass Spectrometer System operated in a MALDI-TOF mode. Absorption spectra were measured on a Milton Roy Spectronic 3000 Array spectrophotometer. Steady-state photoluminescence (PL) spectra were recorded on a Perkin-Elmer spectrofluorometer LS 55. Quantum yield was determined by a Quanta-φ integrating sphere. Circular dichroism (CD) measurements were carried out on an Applied Photophysics Chirascan Plus spectropolarimeter. Circularly polarized luminescence (CPL) spectra of the solid was recorded on JASCO CPL-300 at room temperature.

**High-pressure experiment**. High-pressure experiment was carried out using Sapphire-anvil cell. The culet diameter of the sapphire anvils was 0.6 mm, while copper gasket was preindented to a thickness of 0.17 mm, and center hole of 0.24 mm was drilled for the single crystal. A ruby chip was inserted into the sample compartment for in situ pressure calibration according to the R1 ruby fluorescence method. Silicone oil was used as the pressure-transmitting medium. The hydrostatic pressure on the single crystal was determined through monitoring the widths and separation of the R1 and R2 lines. The PL measurement under high pressure was performed on a QuantaMaster 40 spectrometer in the reflection mode. The 380 nm line of a xenon lamp with a power of 60 W was used as the excitation source. The fluorescence images of the single crystal under visible light and 380 nm UV light were taken by putting the Sapphire-anvil cell (SAC) containing the sample on a Nikon fluorescence microscope. All experiments were conducted at room temperature.

**Microbial imaging**. Prior to bacteria imaging, a single colony of bacteria on solid culture medium [Luria broth (LB) for *E. coli*] was transferred to 50 mL of the liquid culture medium and grown at 37 °C for 12–16 h. The concentrations of bacteria were determined by measuring the optical density at 600 nm (OD600) and then $10^9$ colony forming unit (CFU) of bacteria was transferred to a 1.5 mL EP tube. Bacteria was harvested by centrifuging at 10,000 rpm for 5 min. After removal of the supernatant, 1 mL PBS was added into the EP tube to wash the bacteria, then the bacteria were collected by centrifuging at 10,000 rpm for 5 min again. After removal of the supertant, 1 mL 50 μM dye solution in PBS was added into the EP tube. After dispersing with a vortex, the bacteria were incubated at room temperature for 60 min. To take fluorescence images, about 10 μL of the stained bacteria solution was transferred to a glass slide and then covered by a coverslip. The image was collected using a 100x objective. The bacteria was imaged under a Confocal microscope (Zeiss LSM 710) using 405 nm excitation wavelength.

Prior to fungi imaging, Penicillium chrysogenum was cultured on Malt Extract Agar for 7 days at 28 °C. Gently scrape the surface growth from the culture of the test organism using an inoculating wire. Putting the spore into a sterile container of sterile water, vigorously shaking the flask or sonicate to liberate the spores from the fruiting bodies and to break the spore clumps. Remove large mycelial fragments and clumps of agar. The spore suspension was store at 4 °C. Twenty microliters 1 mM dye stock solution was added into 980 μL *P. chrysogenum* spore suspension and incubated at room temperature for 3 h. To take fluorescence images, about 10 μL of the stained bacteria solution was transferred to a glass slide and then covered by a coverslip. The image was collected using a 100x objective. The fungus was imaged under a Confocal microscope (Zeiss LSM 710) using 405 nm excitation wavelength.

## Data availability
All relevant data that support the findings of this study are available from the corresponding authors upon reasonable request.

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

## Acknowledgements

This work was supported by the National Science Foundation of China (51622305 and 21788102), the Innovation and Technology Commission (ITC-CNERC14SC01 and ITS/254/17), the Research Grants Council of Hong Kong (16305015, 16308016, A-HKUST605/16, C2014–15G, and C6009–17G), the PCSIRT (IRT13023) and the Shenzhen Science and Technology Program (JCYJ20160229205601482 and JCYJ20160509170535223), the UGC Special Equipment Grant (SEG-HKU-07) as well as the University of Hong Kong Development Fund 2013–2014 project New Ultrafast Spectroscopy Experiments for Shared Facilities

## Author contributions

Z.Z. and B.Z.T. conceived and designed the experiments. Z.Z., C.L.L., X.X.G., Y.Y and X.Q.Z performed the synthesis. Z.Z and Y.X. did the PL measurement and analyzed the data. X.Y.Z., Y.J.C., and X.H.H. performed the theoretical calculation. W.H. performed the microbial imaging experiments. F.Y.S and J.Z performed the CPL measurement. L.L.D. and D.L.P. performed the time-resolved spectra and analyzed the data. Y.J.L. and B.X. performed the high-pressure fluorescence measurement and the materials studio simulation. R.T.K.K, J.W.Y.L, X.W.H. and H.W. took part in the discussion and give important suggestions. Z.Z., X.Y.Z., L.L.D., D.L.P., and B.Z.T. co-wrote the paper.

## Additional information

**Competing interests:** The authors declare no competing interests.

