## [Peer Review File · Nature Communications]

Reviewers' comments:

Reviewer #1 (Remarks to the Author):

This manuscript from Tang et al. discloses the aggregation-induced emission behaviour of cyclooctatetraene derivatives. By experimentally and computationally exploring the propensity of the saddle-shaped compounds to invert their conformation in solution and the solid state, the authors infer that solution-state emission is quenched by passing through a Baird aromatic structure. When the accessible conformations are more restricted in the solid state, the materials become more emissive.

The claims made in the manuscript are novel. There has been an extended period of interest in the mechanisms behind and applications of 'aggregation-induced emission' (AIE). Recent time-resolved experiments from the same authors (e.g., Chem. Sci. 2018, 9, 4662) have essentially settled the debate on the behaviour of the simplest and most common AIE emitter, tetraphenylethylene, which relaxes by rotation of hindered single bonds and partial double bonds in the solid state, as well as by photocyclisation. The present system is conceptually quite distinct and, to the best knowledge of this Reviewer, is the first to invoke a Baird aromatic state in the mechanism. Given the broad applicability of AIE materials in OLEDs, cell staining etc., this novel mechanism represents an advance that will, in principle, be of broad interest to the readership of Nature Communications. It is the opinion of this Reviewer, therefore, that the manuscript should be accepted after addressing the issues discussed below.

1. The authors have used several different techniques to investigate the excited state dynamics, including time-resolved measurements, a racemisation study, and computational modelling. This Reviewer commends the rigour of the approach and is convinced that the main claims of the manuscript are supported. However, there does seem to be a subtle gap between the experimental data and the description of the mechanism. The authors propose in the main text and show in Figure S19 that non-radiative decay occurs from the planar transition state structure, although there does not seem to be direct evidence that this is the case. The data show that (i) the planar transition state structure is more accessible in the excited state and that (ii) the non-radiative decay rate is increased in solution, where the same transition state is also more accessible, compared to the solid samples. But this correlation does not necessarily imply causation. Might it be that the non-radiative decay is enhanced in another more accessible conformation, distinct from the planar transition state? Unless the authors can rule out this possibility, this Reviewer would recommend using less definitive language, clearly stating that the mechanism is a hypothesis which fits the data, while other scenarios may be possible.

2. This investigation builds on the significant precedent of Ueda et al. *Nature Commun.* 2017, 8, 346 (reference 29). Ueda et al. reported excited state saddle-inversion through a Baird aromatic structure for the same cyclooctatetraene that features prominently in the present manuscript. Ueda et al. did not report on the AIE characteristics of the cyclooctatetraene, so the present manuscript constitutes a significant advance. In the opinion of this Reviewer, however, there should be more explicit statements in the manuscript about where there is overlap, replacing the slightly vague descriptions used in the submitted draft. The experiments shown in Figure 4, for example, are essentially repeats of measurements reported by Ueda et al., which should be made clear to the reader.

3. The readability of the manuscript would be enhanced significantly by including labelled diagrams of the chemical structures in the main text. It is unreasonable to expect readers to refer to figure captions (or even the SI!) to confirm the structures of COTh, COTh-TMS and COTh-Py. The discussion of the synthetic strategy in the main text, referring to intermediates 1–3, cannot be understood at present without referring to the SI.

4. PLQYs are reported but there does not appear to be information about how they were measured. These experimental details should be added.

5. On line 205, the statement 'significant enhancement of both k_r and k_{ic} ' is confusing as one rate is increased while the other decreases.

6. Details of the concentrations, temperature, and solvents used should be included for Figure 4c–f. Can the authors be sure that the temperature of the irradiated samples does not increase, which could contribute to the increased rate of racemisation?

7. Out of curiosity, does the CPL signal of the enantioenriched crystals change when irradiated? Similarly, could the enantiomeric excess of a sample (solid or solution) be enhanced by irradiation with circularly polarised light? These experiments would not necessarily impact the present manuscript, but may be worth investigating.

8. The English language quality and presentation of the manuscript are generally sound but require some attention before publication. A recurring error is the spelling of 'Baird' (written as 'Barid' here), which must be corrected.

9. The intensity of ^1H NMR signals should be corrected to match the number of magnetically equivalent nuclei in the compounds. For example, the signal at 7.54 ppm for compound 3 corresponds to 8H rather than 2H.

10. No ^{13}C NMR data are included for compound 3. These data should be included.

11. The high-resolution mass data for COT h -Py requires some attention. The procedure states a m/z of 1688.5, but Fig. S35 states 1661.6. Moreover, the peaks selected, $[\text{M}+\text{Na}]^+$ or $[\text{M}]^+$, are difficult to distinguish from noise and unlikely to be the most prominent ions. This Reviewer suggests double checking the data and, in particular, looking for a signal corresponding to $[\text{M}-\text{Br}]^+$ or other similar ions, which are likely to be present for a salt such as COT h -Py.

12. Details of the solvent used for UV-vis should be added to Figure S3.

Reviewer #2 (Remarks to the Author):

In this work, the luminescent behavior of thiophene-fused COT and its derivatives was investigated by photoluminescence (PL) spectra, time-resolved absorption spectra, circular dichroism (CD) and circular polarized luminescence (CPL) spectroscopy as well as pressure-dependent fluorescent spectra. The paper was clearly written and the compounds were well characterized. All the spectra data are reliable.

This work is very important, because it developed and demonstrated a new family of non-aromatic annulene as AIE materials triggered by aromaticity reversal. In my opinion it could be suitable for publication in Nature Communication. Nonetheless, I do have a few suggestions that I feel would improve the paper:

1. The authors should further prove the aromaticity of the excited-state (and the non-aromaticity of ground-state for comparison) because using the 4N rule plus Baird's rule to judge the aromaticity of the studied molecules is insufficient. The accepted criteria to judge the aromaticity include NICS, AICD etc., which can be easily calculated by DFT study.

2. Page 14 line 261. There should be a space in ".....betweenthe....."

3. Page 22 line 415. Reference 23 does not have a page number.

Reviewer #1 (Remarks to the Author):

This manuscript from Tang et al. discloses the aggregation-induced emission behaviour of cyclooctatetraene derivatives. By experimentally and computationally exploring the propensity of the saddle-shaped compounds to invert their conformation in solution and the solid state, the authors infer that solution-state emission is quenched by passing through a Baird aromatic structure. When the accessible conformations are more restricted in the solid state, the materials become more emissive.

The claims made in the manuscript are novel. There has been an extended period of interest in the mechanisms behind and applications of ‘aggregation-induced emission’ (AIE). Recent time-resolved experiments from the same authors (e.g., Chem. Sci. 2018, 9, 4662) have essentially settled the debate on the behaviour of the simplest and most common AIE emitter, tetraphenylethylene, which relaxes by rotation of hindered single bonds and partial double bonds in the solid state, as well as by photocyclisation. The present system is conceptually quite distinct and, to the best knowledge of this Reviewer, is the first to invoke a Baird aromatic state in the mechanism. Given the broad applicability of AIE materials in OLEDs, cell staining etc., this novel mechanism represents an advance that will, in principle, be of broad interest to the readership of Nature Communications. It is the opinion of this Reviewer, therefore, that the manuscript should be accepted after addressing the issues

Response: We sincerely thank the appreciation and recognition of the reviewer on our work. We have provided a detailed point-to-point response to each question.

1. The authors have used several different techniques to investigate the excited state dynamics, including time-resolved measurements, a racemisation study, and computational modelling. This Reviewer commends the rigour of the approach and is convinced that the main claims of the manuscript are supported. However, there does seem to be a subtle gap between the experimental data and the description of the

mechanism. The authors propose in the main text and show in Figure S19 that non-radiative decay occurs from the planar transition state structure, although there does not seem to be direct evidence that this is the case. The data show that (i) the planar transition state structure is more accessible in the excited state and that (ii) the non-radiative decay rate is increased in solution, where the same transition state is also more accessible, compared to the solid samples. But this correlation does not necessarily imply causation. Might it be that the non-radiative decay is enhanced in another more accessible conformation, distinct from the planar transition state? Unless the authors can rule out this possibility, this Reviewer would recommend using less definitive language, clearly stating that the mechanism is a hypothesis which fits the data, while other scenarios may be possible.

Response: We thank the reviewer for the extremely professional comments to improve our manuscript. In the last manuscript, we proposed the non-radiative decay mainly occur through the planar transition state structure because it matches well with the chirality racemization of the CD experiment. However, after carefully consider the reviewer's suggestion and read the paper of Ueda and co-workers (Nat. Commun. 2017, 8, 346), we realized that the racemization can also easily happen in the excited state since the two minimum energy structure can easily transform with each other by overcoming the energy barrier between the minimum energy structure and the transition state structure. In that case, the non-radiative decay through the minimum energy structure should be favored. Therefore, as the reviewer mentioned, the non-radiative decay possibly was enhanced in other more accessible conformations such as the lowest excited state besides the planar transition state, and such claim is also consistent well with the CD experimental results. After carefully consideration of the reviewer's suggestions, we correct the hypothesis as follow and supplement the corresponding discussion in the revised manuscript.

Supplementary Figure S34. a. Proposed decay pathways along the potential energy surface of COTh in dilute solution. **b.** Proposed decay pathways along the potential energy surface of COTh in the solid state. Abbreviation: GS = ground state; ES = excited state; TS = transition state; MES = minimum energy structure; A = absorption; F = fluorescence; NR = Non-radiative decay.

2. This investigation builds on the significant precedent of Ueda et al. *Nature Commun.* 2017, 8, 346 (reference 29). Ueda et al. reported excited state saddle-inversion though a Baird aromatic structure for the same cyclooctatetraene that features prominently in the present manuscript. Ueda et al. did not report on the AIE characteristics of the cyclooctatetraene, so the present manuscript constitutes a significant advance. In the opinion of this Reviewer, however, there should be more explicit statements in the manuscript about where there is overlap, replacing the slightly vague descriptions used in the submitted draft. The experiments shown in Figure 4, for example, are essentially repeats of measurements reported by Ueda et al., which should be made clear to the reader.

Response: Thanks for the suggestion of the reviewer, we agree with reviewer that the work of Ueda et al. gives us important clues in explaining the AIE mechanism of COTh system. And the integration of Baird aromaticity with AIE is of high novelty. In the revised manuscript, we highlighted the work of Ueda and coworkers with the comment of “Indeed, the pioneering work by Itoh and Hada et al., indicates some

heterocyclic fused COT underwent a large magnitude of “up-down” conformation inversion upon photoexcitation.” and gave explicit statement as “It is worthy to note that this experiment is exact the same measurements performed previously by Itoh et al., in which they investigated the conformation change of 2-methylthiophene fused COT, a structural analogue of COTh, via CD spectroscopy.”

3. The readability of the manuscript would be enhanced significantly by including labelled diagrams of the chemical structures in the main text. It is unreasonable to expect readers to refer to figure captions (or even the SI!) to confirm the structures of COTh, COTh-TMS and COTh-Py. The discussion of the synthetic strategy in the main text, referring to intermediates 1–3, cannot be understood at present without referring to the SI

Response: To improve the readability of the manuscript, we have added the molecular structures involved in the study and the synthetic route to compound COTh-Py in Figure 2 as follow.

Figure 2. The structures of COTh and its derivatives and the synthetic route to COTh-Py.

4. PLQYs are reported but there does not appear to be information about how they were measured. These experimental details should be added.

Response: The PLQY was measured by using a calibrated integrating sphere. We have added the experimental details in the revised manuscript and supplementary information.

5. On line 205, the statement 'significant enhancement of both k_r and k_{ic} ' is confusing as one rate is increased while the other decreases.

Response: We are sorry that we have made a typo mistake. From supplementary

Table S3 we can see that from the gas phase to the solid state, the k_r increased while the k_{ic} decreased. We have changed the description as “It is demonstrated that the fluorescence quantum yield of COTh in the solid state is three-orders of magnitude larger than that in the isolated state, because of the significant enhancement of k_r and decrease of k_{ic} .”

Supplementary Table S3. Calculated radiative rate (k_r), non-radiative rate (k_{ic}) and the quantum yield (η_F).

	k_r (10^6 s $^{-1}$)	k_{ic} (10^6 s $^{-1}$)	η_F (%)
gas phase	0.098	4130	0.002
solid phase	0.605	18.7	3.13

6. *Details of the concentrations, temperature, and solvents used should be included for Figure 4c–f. Can the authors be sure that the temperature of the irradiated samples does not increase, which could contribute to the increased rate of racemisation?*

Response: The concentration of the solutions was fixed as 10 μ M and the measurement is carried out at laboratory with air-condition and stable temperature. Freshly distilled THF was used as solvent for the CD experiment. We have added all the experimental details in Figure 4c–f. In addition, we used a thermographic camera to monitor the temperature change of the UV irradiated sample. Results indicate that during the UV lamp irradiation (up to 200 s), the temperature of the sample solution keeps almost unchanged. Thus, the temperature effect on the racemization process can be excluded.

7. *Out of curiosity, does the CPL signal of the enantioenriched crystals change when irradiated? Similarly, could the enantiomeric excess of a sample (solid or solution) be enhanced by irradiation with circularly polarised light? These experiments would not necessarily impact the present manuscript, but may be worth investigating.*

Response: Thanks very much for the valuable suggestions of the reviewer. We have investigated the change of the CPL signal of the enantioenriched crystals upon UV irradiation with the same sample preparation method and same UV source as the CD experiment. Results show that the CPL signal of the enantioenriched crystals remains almost unchanged upon UV irradiation for 14 min, which is ascribed to the restricted molecular conformation inversion in the crystals. Additionally, investigation of the change of enantiomeric excess of a sample under the irradiation of circularly polarized light is interesting. Unfortunately, it is hard for us to do the measurement currently since we can't get a suitable polarized plate for UV light source. We think it should be a very interesting project that worths further study. We are looking for collaborators that can perform such study in our next project.

Supplementary Figure S18. g_{PL} signal of COTh (+) and COTh (-) in the solid state upon UV lamp irradiation. **a.** g_{PL} spectra of COTh (+) under UV irradiation. **b.** The plot of g_{PL} value of COTh (+) versus irradiation time. **c.** g_{PL} spectra of COTh (-) under UV irradiation. **d.** The plot of g_{PL} value of COTh (-) versus irradiation time. Excitation wavelength: 360 nm. The irradiation wavelength of UV lamp: 254 nm.

8. *The English language quality and presentation of the manuscript are generally sound but require some attention before publication. A recurring error is the spelling of 'Baird' (written as 'Barid' here), which must be corrected.*

Response: We are sorry for such careless mistakes. The manuscript has been double-checked and we have corrected "Barid" as "Baird" in the revised manuscript.

9. *The intensity of 1H NMR signals should be corrected to match the number of magnetically equivalent nuclei in the compounds. For example, the signal at 7.54 ppm for compound 3 corresponds to 8H rather than 2H.*

Response: Thanks for the suggestion of the reviewer, we have corrected the number of H signals in the revised supplementary information as follow: ^1H NMR (400 MHz, CDCl_3 , 298k), δ (ppm): 7.54–7.51 (d, $J = 8$ Hz, 8H), 7.13 (s, 4H), 6.93–6.91 (d, $J = 8$ Hz, 8H), 3.84 (s, 12H), which is matched well with the number of magnetically equivalent nuclei in the compound.

10. No ^{13}C NMR data are included for compound 3. These data should be included.

Response: We have supplemented the ^{13}C NMR spectrum and the corresponding data of compound 3 in the revised supplementary information as follow.

^{13}C NMR (150 MHz, $\text{C}_2\text{D}_2\text{Cl}_4$, 298k), δ (ppm): 160.9, 146.6, 138.8, 131.9, 128.3, 127.7, 126.3, 115.9, 56.9.

Supplementary Figure S29. ^1H NMR of compound 3 in 1,1,2,2-tetrachloroethane- d_2 at 298K (150 MHz).

11. The high-resolution mass data for COTh-Py requires some attention. The procedure states a m/z of 1688.5, but Fig. S35 states 1661.6. Moreover, the peaks selected, $[M+Na]^+$ or $[M]^+$, are difficult to distinguish from noise and unlikely to be the most prominent ions. This Reviewer suggests double checking the data and, in particular, looking for a signal corresponding to $[M-Br]^+$ or other similar ions, which are likely to be present for a salt such as COTh-Py.

Response: Thanks for the suggestion of the reviewer, we have repeated the HRMS measurement and found the signal corresponding to $[M-4Br]^{4+}$. We have changed Figure S35 in the revised supplementary information.

Supplementary Figure S40. High resolution mass spectrum of compound COTh-Py.

12. Details of the solvent used for UV-vis should be added to Figure S3.

Response: The solvent used to obtain the UV-vis spectrum shown in Figure S3 is distilled tetrahydrofuran (THF). We have added the details in the revised Supplementary information.

Reviewer #2 (Remarks to the Author):

In this work, the luminescent behavior of thiophene-fused COT and its derivatives was

investigated by photoluminescence (PL) spectra, time-resolved absorption spectra, circular dichroism (CD) and circular polarized luminescence (CPL) spectroscopy as well as pressure-dependent fluorescent spectra. The paper was clearly written and the compounds were well characterized. All the spectra data are reliable.

Response: Thanks very much for the appreciation and recognition of the reviewer on our work. We have provided a detailed point-to-point response to each question.

This work is very important, because it developed and demonstrated a new family of non-aromatic annulene as AIE materials triggered by aromaticity reversal. In my opinion it could be suitable for publication in Nature Communication. Nonetheless, I do have a few suggestions that I feel would improve the paper:

1. The authors should further prove the aromaticity of the excited-state (and the non-aromaticity of ground-state for comparison) because using the $4N$ rule plus Baird's rule to judge the aromaticity of the studied molecules is insufficient. The accepted criteria to judge the aromaticity include NICS, AICD etc., which can be easily calculated by DFT study.

Response: Thanks very much for the suggestion of the reviewer. We have supplemented calculation data to confirm the excited-state aromaticity of COTh and the results are present as follow. The results indicate that for both optimized minimum energy structures and the transition state structures, the excited states show better planarity and smaller bond length difference than the ground state, suggesting the aromatic characteristic of the excited state (Supplementary Figure S19 and S20). The NICS_{zz} scan along the orthogonal direction of the central ring of COTh showed positive and negative chemical shifts for the transition state structures in the S0 and T1 states, respectively, which is consistent with the ground-state antiaromaticity and excited-state aromaticity of COTh (Supplementary Figure S21). Furthermore, ACID for the transition state structures in the S0 and T1 states show counterclockwise and clockwise ring current, corresponding to the ground-state antiaromaticity and

excited-state aromaticity characteristic of COTh (Supplementary Figure S22). We have added these data in the revised supplementary information.

Supplementary Figure S19. The optimized minimum energy structure and the bond lengths of COTh in S_0 , S_1 and T_1 . **a.** The optimized molecular structures of COTh in S_0 , S_1 and T_1 . **b.** The chemical structure of COTh with atoms in the central ring labeled. **c.** Bond lengths of the central ring of COTh in S_0 , S_1 and T_1 .

Supplementary Figure S20. The optimized transition state structure and the bond lengths of COTh in S_0 , S_1 and T_1 . **a.** The optimized transition state structures of COTh in S_0 , S_1 and T_1 . **b.** The chemical structure of COTh with atoms in the central ring labeled. **c.** Bond lengths of the central ring of COTh in S_0 , S_1 and T_1 .

Supplementary Figure S21. Nucleus-independent chemical shift (NICS) scans of COTh of the transition state structures in the S_0 and T_1 states. **a.** NICS_{zz} scans start from the center of the central ring and scanned along the arrow. **b.** The curves

recorded by NICSzz scan, black- and green-colored curves represent the scans in the S_0 and T_1 states, respectively

Supplementary Figure S22. Anisotropy of the induced current density (ACID) isosurfaces of the transition state structures in the S_0 and T_1 state. Arrows on the left mean the direction of the ring current.

2. Page 14 line 261. There should be a space in ".....betweenthe....."

Response: We have corrected it in the revised manuscript.

3. Page 22 line 415. Reference 23 does not have a page number.

Response: We have added a page number for reference 23

Reviewer #1 (Remarks to the Author):

The authors have fully addressed the Reviewers' comments by performing new experiments and calculations, as well as amending figures and text. In the opinion of this reviewer, all of the issues raised have been addressed satisfactorily except one minor error that remains (see below). The manuscript is suitable for acceptance.

- The updated HRMS data for COTh-Py still appears to be incorrect. The peak shown at $m/z = 1344.5666$ does not correspond to $[M-4Br]4+$ as stated. This ion would instead appear as a peak at $m/z = 322$. The peaks expected for loss of counterions would be at $m/z = 1527 [M-Br]^+$; $724 [M-2Br]2+$; $456 [M-3Br]3+$; $322 [M-4Br]4+$. The authors should recheck the MS data and, if necessary, use a different technique to confirm the molecular formula.

Reviewer #2 (Remarks to the Author):

All the question have been fully addressed, thus, it is suitable for publication in Nature Communication.

Response to referees

Reviewer #1 (Remarks to the Author):

The authors have fully addressed the Reviewers' comments by performing new experiments and calculations, as well as amending figures and text. In the opinion of this reviewer, all of the issues raised have been addressed satisfactorily except one minor error that remains (see below). The manuscript is suitable for acceptance.

The updated HRMS data for COTh-Py still appears to be incorrect. The peak shown at $m/z = 1344.5666$ does not correspond to $[M-4Br]4+$ as stated. This ion would instead appear as a peak at $m/z = 322$. The peaks expected for loss of counterions would be at $m/z = 1527 [M-Br]^+$; $724 [M-2Br]2+$; $456 [M-3Br]3+$; $322 [M-4Br]4+$. The authors should recheck the MS data and, if necessary, use a different technique to confirm the molecular formula.

Response: Thanks for the suggestion of the reviewer, we have conducted the MS measurement by using ESI ionization method, which successfully detected the peaks of $[M-2Br]^{2+}$, $[M-3Br]^{3+}$, and $[M-4Br]^{4+}$. The MS information has been updated in the supplementary information as follow: HRMS (ESI): m/z : $[M-4Br]^+$ calcd for $C_{84}H_{88}N_4O_4S_4$, 336.5; found, 336.6; $[M-3Br]^+$ calcd for $C_{84}H_{88}N_4O_4S_4Br$, 475.2; found, 475.1; $[M-2Br]^+$ calcd for $C_{84}H_{88}N_4O_4S_4Br_2$, 752.8; found, 752.4.

Supplementary Figure 13. Mass spectrum of compound **COTh-Py**.

Reviewer #2 (Remarks to the Author):

All the question have been fully addressed, thus, it is suitable for publication in Nature Communication.

Response: We thank the recognition of the reviewer for our revision.